# A stochastic numerical approach for a class of singular singularly perturbed system

**Zulqurnain Sabir**[1], **Thongchai Botmart**[2]*, **Muhammad Asif Zahoor Raja**[3], **Wajaree Weera**[2], **Fevzi Erdoğan**[4]

**1** Department of Mathematics and Statistics, Hazara University, Mansehra, Pakistan, **2** Department of Mathematics, Faculty of Science, Khon Kaen University, Khon Kaen, Thailand, **3** Future Technology Research Center, National Yunlin University of Science and Technology, Douliou, Yunlin, Taiwan, R.O.C, **4** Department of Mathematics, Faculty of Sciences, Yuzuncu Yil University, Van, Turkey

* thongbo@kku.ac.th

**Data Availability Statement:** All relevant data are within the paper.

**Funding:** This research received funding support from the NSRF via the Program Management Unit

## Abstract

In the present study, a neuro-evolutionary scheme is presented for solving a class of singular singularly perturbed boundary value problems (SSP-BVPs) by manipulating the strength of feed-forward artificial neural networks (ANNs), global search particle swarm optimization (PSO) and local search interior-point algorithm (IPA), i.e., ANNs-PSO-IPA. An error-based fitness function is designed using the differential form of the SSP-BVPs and its boundary conditions. The optimization of this fitness function is performed by using the computing capabilities of ANNs-PSO-IPA. Four cases of two SSP systems are tested to confirm the performance of the suggested ANNs-PSO-IPA. The correctness of the scheme is observed by using the comparison of the proposed and the exact solutions. The performance indices through different statistical operators are also provided to solve the SSP-BVPs using the proposed ANNs-PSO-IPA. Moreover, the reliability of the scheme is observed by taking hundred independent executions and different statistical performances have been provided for solving the SSP-BVPs to check the convergence, robustness and accuracy.

## 1. Introduction

The two-point singular singularly perturbed boundary value problems (SSP-BVPs) have numerous applications, such as quantum mechanics, fluid dynamics, the theory of optimal control, geophysics, theory of chemical reactor, elasticity, aerodynamics and gas porous electrode theory [1, 2]. The SSP-BVPs are considered difficult and grim to solve due to the perturbation and singular singularly nature. There are only a few methods available in the literature to handle these types of equations, which involve the perturbation, singularly nature and singularity. Schmeiser et al [3] provided the numerical and asymptotic techniques to solve the SSP-BVPs. Ascher [4] discussed a symmetric difference scheme for solving SSP-BVPs. Mohanty and Arora [5] provided the numerical solutions of these problems using the methods of non-uniform mesh tension spline and convergent tension spline. Zhu [6] presented the asymptotic results assembled by the modified Vasil-eva scheme. He also proved the uniqueness, the existence of the exact solution and the uniform strength of the traditional asymptotic

for Human Resources & Institutional Development, Research and Innovation (grant number B05F640088). The funders had a role in study design, data collection, analysis, and decision to publish.

**Competing interests:** The authors have declared that no competing interests exist.

solution. Kadalbajoo and Aggarwal [7] proposed the B-spline technique for solving the SSP-BVPs. Rashidnia et al. [8] provided the kernel space and cubic spline approaches for solving the SSP-BVPs.

All the above techniques have their individual efficiency, applicability, exactness and flaws over one another. However, the stochastic solvers have never been applied to solve the SSP-BVP by using the artificial neural networks (ANNs) along with global search particle swarm optimization (PSO) and local search interior-point algorithm (IPA), i.e., ANNs-PSO-IPA. The artificial neural networks have been implemented to solve a variety of different applications; some recent applications of the stochastic solvers are circuit theory, higher order singular model, fuel ignition model, induction of the motor models, Thomas-Fermi model, doubly singular nonlinear systems, nanotechnology, nanofluidics, chaos control of Bonhoeffer–van der Pol, nonlinear equations, Troesch's problem, controls, communication systems, particle physics, linear and nonlinear fractional order model, physical models signified nonlinear system of equations, financial mathematics, multiple singularities models based on Painleve equations etc., see [9–13] and references cited therein. Keeping view of these facts, authors are inspired to propose new computing criteria through the ANNs modelling. Therefore, the stochastic ANNs-PSO-IPA procedures are proposed to solve the SSP-BVPs. The general form of SSP-BVPs is given as [1]:

$$\varepsilon \frac{d^2 y}{dt^2} + \frac{p(t)}{t} \frac{dy}{dt} + \frac{q(t)}{t^2} y(t) = g(t), \quad t \in (0,1)$$
$$y(0) = A, \ y(1) = B, \quad A, B \in \Re,$$

(1)

where $\varepsilon$ is a positive small parameter, $0 < \varepsilon << 1$, $p(t) \geq$ and $q(t), g(t)$ smoothly used to satisfy the uniqueness, existence of the solution constants.

Some salient features of the ANNs-PSO-IPA for solving the SSP-BVPs are presented as:

- Exploitation and investigation of stochastic ANNs-PSO-IPA solvers to determine the accurate, consistent and reliable numerical outcomes of the SSP-BVPs.

- The designed technique is implemented efficiently for two problems with various cases of the SSP-BVPs that demonstrates the proficiency of the designed approach.

- The correctness of the proposed numerical ANNs-PSO-IPA is observed by using the comparison procedures for solving the SSP-BVPs.

- The proposed scheme is effective through the statistical procedures of different performances based on the mean, standard deviation and root mean square errors.

- The SSP-BVPs are not easy to solve due to the complicated behavior. Therefore, ANNs is a better choice to handle such kinds of complex problems, as well as it solves other challenging mathematical, biological, engineering and physical models for which the traditional methodologies do not work.

## 2. Mathematical model to solve singular singularly perturbed model

In the first phase, the structure of the SSP-BVPs is presented using an unsupervised error-based fitness function. The unidentified weight vectors are trained using the ANNs-PSO-IPA is studied in the second phase. In Fig 1, the graphical abstract of the system model is presented.

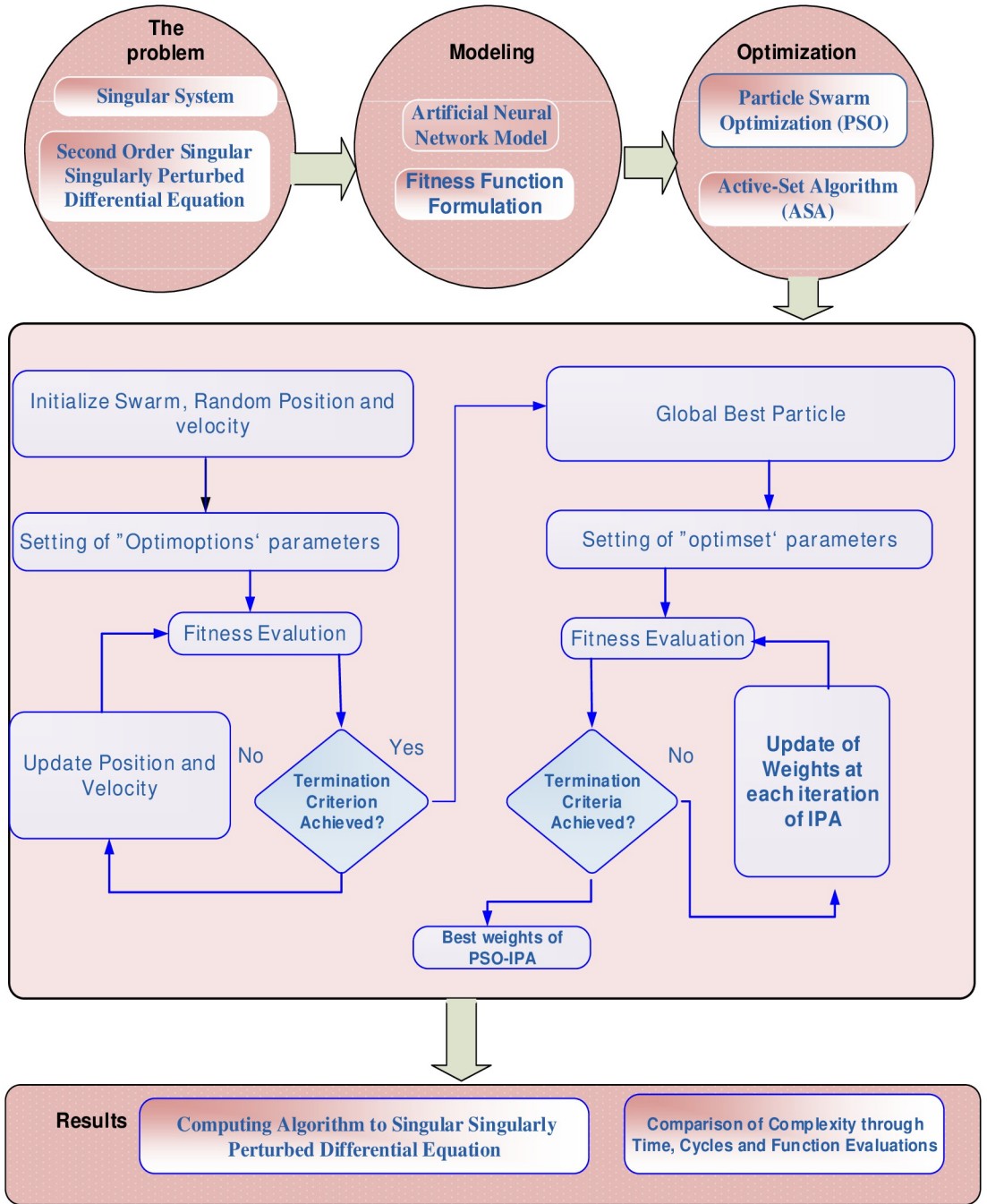

**Fig 1. Graphical abstract of suggested scheme for SSP-BVPs.**

## 2.1 ANNs modeling

The mathematical systems are assembled for the SSP-BVPs with the feed-forward ANNs strength having continuous mapping form through the single input, hidden and outer layers

for the proposed outcomes $u(t)$ is given as:

$$
\begin{aligned}
\hat{u}(t) &= \sum_{j=1}^{k} \alpha_j f(\xi_j t + \beta_j), \\
\frac{d\hat{u}}{dt} &= \sum_{j=1}^{k} \alpha_j \frac{d}{dt} f(\xi_j t + \beta_j), \\
\frac{d^2 \hat{u}}{dt^2} &= \sum_{j=1}^{k} \alpha_j \frac{d^2}{dt^2} f(\xi_j t + \beta_j), \\
&\quad \vdots \\
\frac{d^{(l)} \hat{u}}{dt^{(l)}} &= \sum_{j=1}^{k} \alpha_j \frac{d^{(l)}}{dt^{(l)}} f(\xi_j t + \beta_j).
\end{aligned}
\tag{2}
$$

Where, the log-sigmoid $f(t) = 1/(1 + e^{-t})$ function is applied as an activation function. The updated form of Eq (2) becomes as:

$$
\begin{aligned}
\hat{u}(t) &= \sum_{j=1}^{k} \alpha_j \left( \frac{1}{1 + e^{-(\xi_j t + \beta_j)}} \right), \\
\frac{d\hat{u}}{dt} &= \sum_{j=1}^{k} \alpha_j \xi_j \left( \frac{e^{-(\xi_j t + \beta_j)}}{\left(1 + e^{-(\xi_j t + \beta_j)}\right)^2} \right), \\
\frac{d^2 \hat{u}}{dt^2} &= \sum_{j=1}^{k} \alpha_j \xi_j^2 \left( \frac{2 e^{-2(\xi_j t + \beta_j)}}{\left(1 + e^{-(\xi_j t + \beta_j)}\right)^3} - \frac{e^{-(\xi_j t + \beta_j)}}{\left(1 + e^{-(\xi_j t + \beta_j)}\right)^2} \right), \\
&\quad \vdots \\
\frac{d^{(l)} \hat{u}}{dt^{(l)}} &= \sum_{j=1}^{k} \alpha_j \xi_j^{(l)} \left( \frac{2 e^{-n(\xi_j t + \beta_j)}}{\left(1 + e^{-(\xi_j t + \beta_j)}\right)^{l+n}} - \frac{e^{-(n-l)(\xi_j t + \beta_j)}}{\left(1 + e^{-(\xi_j t + \beta_j)}\right)^{n}} \right).
\end{aligned}
\tag{3}
$$

## 2.2 Formulation of fitness function

The appropriate grouping of the Eq (3) is applied to formulate the SSP-BVPs. An objective/ merit function is expressed for SSP-BVP to define an error-based functions as:

$$
\in = \in_1 + \in_2
\tag{4}
$$

Where, $\in_1$ is the merit functions related to differential model as:

$$
\in_1 = \frac{1}{N} \sum_{l=1}^{N} \left( t_m^2 \varepsilon \frac{d^2 \hat{y}_m}{dt^2} + t_m p_m \frac{d\hat{y}_m}{dt} + q_m \hat{y}_m - t_m^2 g_m \right)^2
\tag{5}
$$

$$
N = \frac{1}{h}, \ \hat{y}_m = \hat{y}(t_m), \ p_m = p(t_m), \ q_m = q(t_m), \ g_m = g(t_m) \text{ and } t_m = mh.
$$

Where $h$ is the step size. Likewise, $\in_2$ signifies an error function associated to the BCs, written as:

$$
\in_2 = \frac{1}{2} \left( (\hat{y}_0 - A)^2 + (\hat{y}_N - B)^2 \right).
\tag{6}
$$

## 2.3 Hybrid computing PSO-IPA

Design parameters or unknown ANNs weights are modified using the hybrid computing background of the PSO reinforced with IPA. PSO is used as a replacement to genetic algorithms [14] and become most commonly choice using the optimization performances and minor memory requirements [15]. PSO is used to cooperate the swarm performances of birds flocking as well as fish schooling [16]. Recently, PSO is used to predict the moisture contents of poplar fibers [17], communication systems [18], solar photovoltaic system [19], freeway ramp metering [20], real-time measurement of microgrid islanding [21], formulation of computer model with economic measures [22] and parameter identification with control [23]. Every particle contains the fitness performances describing the problem standards is known as merit function. PSO provides optimal results iteratively to initialize the parameter runs. The velocity and position using the recognized local positions are represented as $\boldsymbol{P}_{LB}^{r-1}$ and global best positions are denoting as $\boldsymbol{P}_{GB}^{r-1}$. The updated position and velocity form is provided as:

$$X_j^n = X_j^{n-1} + V_j^{n-1} \tag{7}$$

$$V_j^n = \omega V_j^{n-1} + b_1 r_1 \left( \boldsymbol{P}_{LB}^{n-1} - X_j^{j-1} \right) + b_2 r_2 \left( \boldsymbol{P}_{GB}^{n-1} - X_j^{n-1} \right) \tag{8}$$

Where the vector $\boldsymbol{X}_j$ and $\boldsymbol{V}_j$ represent the j$^{th}$ particle of swarm and the velocity. $\boldsymbol{r}_1$ and $\boldsymbol{r}_2$ are called random vectors, $\omega \in [0, 1]$ shows the constant of inertia weight. The velocity vector lies in $[-v_{max}, v_{max}]$, $v_{max}$ represents maximum value of the velocity. The upgraded performance of the PSO is pragmatic by hybridization with efficient local search method (LSM) generally presented in the optimization. Therefore, optimization applications-based IPA applied in the replication for convergence to calculate the best optimization of PSO using the initial values of IPA. The workflow of PSO-IPA for SSP-BVPs is portrayed in Fig 1, the parameter settings is provided in Table 1, while the detailed pseudocode of PSO-IPA for solving SSP-BVPs is provided in detailed in Table 2.

## 2.4 Parameters of PSO and IPA

The parameter setting based on the PSO and IPA is provided in Table 1.

**Table 1. Settings of the parameters using PSO and IPA.**

| PSO | | IPA | |
|---|---|---|---|
| **Parameters** | **Values** | **Parameters** | **Values** |
| **Swarm Size** | 70 | **Algorithm** | Active-set |
| **Weights** | Linear decreasing | **Individual Size** | 30 |
| **Particle Size** | 30 | **Max Iterations** | 800 |
| **TolFun** | $10^{-18}$ | **Tol Fun** | 1e$^{-18}$ |
| **Local acceleration** | Linear decreasing | **"TolCon"** | 1e$^{-20}$ |
| **Global acceleration** | Linear Increasing | **Fun evaluations** | 200000 |
| **Population Span** | (-30,30) | **Initial Weights** | Global best of GAs |
| **HybridFcn** | @fmincon | **'TolX'** | 1e$^{-18}$ |
| **Velocity span** | (-2,2) | **Other** | Defaults |

**Table 2. Pseudocode of PSO-IPA to solve the SSP-BVPs.**

```
Start of PSO
  Step 1: {Initialization}
    Randomly generate the particle's initial swarm. Adjust the 'PSO'
    Parameter and 'optimoptions' procedures.
Step 2: {Calculation of Fitness}
  Compute the fitness value for every particle using Eq (4).
Step 3: {Ranking}
  Rank every particle based on the fitness function
Step 4: {Criteria of Stoppage}
  Stop the procedure of optimization for any of the following
    • Achieve the chosen level of fitness
    • Desired Number of flights/cycles achieved
  If ending criteria gotten, then move to Step 5
Step 5: {Renewal}
  Call the position for Eq (7) and velocity using Eq (8).
Step 6: {Improvement}
Repeat the procedure from second to sixth step, until the entire flights are
attained
  Step 7: {Storage}
Store the best values of the fitness and represent it the best global particle
End of PSO
Start of IPA
Step 1: Check the terminating standards, if it meets then terminate the further
process.
Step 2: If stopping standards does not obtained then go back to fourth step.
End of IPA
Start of Statistics
```

## 3. Statistical performance measures

In this study, the statistical performance measure of SSP-BVPs is presented for all cases of both problems using the designed methodology. The performances of the three measures are implemented on the basis of mean absolute deviation (MAD), Theil's inequality coefficient (TIC) operator and Nash Sutcliffe efficiency (NSE) operator. The global Global MAD, Global TIC and Global NSE are applied to solve the SSP-BVPs. The mathematical forms of these operators are given as:

$$\text{MAD} = \frac{1}{n}\sum_{m=1}^{n}|y_m - \hat{y}_m|, \tag{9}$$

$$\text{TIC} = \frac{\sqrt{\frac{1}{n}\sum_{m=1}^{n}\left(y_m - \hat{y}_m\right)^2}}{\left(\sqrt{\frac{1}{n}\sum_{m=1}^{n}y_m^2} + \sqrt{\frac{1}{n}\sum_{m=1}^{n}\hat{y}_m^2}\right)}, \tag{10}$$

$$\text{NSE} = \begin{cases} 1 - \dfrac{\displaystyle\sum_{m=1}^{n} (y_m - \hat{y}_m)^2}{\displaystyle\sum_{m=1}^{n} (y_m - \bar{y}_m)^2}, & \bar{y}_m = \dfrac{1}{n}\sum_{m=1}^{n} y_m, \end{cases} \tag{11}$$

$$\text{E}_{NSE} = 1 - \text{NSE}. \tag{12}$$

Where $n$ is used as an input grid points. In the perfect model case, the values of the above statistical operators are zero.

## 4. Simulations and results

The numerical values are presented here for solving two singular singularly perturbed system of second order ODEs. Moreover, detailed statistical performances are also presented. Two problems along with four cases are demonstrated the accurateness, convergence and efficiency of the proposed scheme. For comparison, the exact results are provided for both of the problems. The term $C$ is used to represent the cases.

### Problem 1: Multiple singular singularly perturbed system

Consider the SSP-BVP of second order is [1]

$$\varepsilon \frac{d^2y}{dt^2} + \frac{1}{t}\frac{dy}{dt} + \frac{1}{t^2}y(t) = 6\varepsilon t - 12\varepsilon t^2 + 4t - 5t^2$$
$$y(0) = 0, \ y(1) = 0 \tag{13}$$

The exact solution of the above Eq (13) is given as

$$y(t) = t^3 - t^4 \tag{14}$$

Four different cases of SSP-BVP (13) are chosen for taking small perturbation values, i.e., $\varepsilon = 2^{-4}, 2^{-6}, 2^{-8}$ and $2^{-10}$. The updated form of the Eq (13) for these four cases based on the small parameters of $\varepsilon$ is given as:

$$2^{-4}\frac{d^2y}{dt^2} + \frac{1}{t}\frac{dy}{dt} + \frac{1}{t^2}y(t) = 6(2^{-4})t - 12(2^{-4})t^2 + 4t - 5t^2,$$

$$2^{-6}\frac{d^2y}{dt^2} + \frac{1}{t}\frac{dy}{dt} + \frac{1}{t^2}y(t) = 6(2^{-6})t - 12(2^{-6})t^2 + 4t - 5t^2,$$

$$2^{-8}\frac{d^2y}{dt^2} + \frac{1}{t}\frac{dy}{dt} + \frac{1}{t^2}y(t) = 6(2^{-8})t - 12(2^{-8})t^2 + 4t - 5t^2,$$

$$2^{-10}\frac{d^2y}{dt^2} + \frac{1}{t}\frac{dy}{dt} + \frac{1}{t^2}y(t) = 6(2^{-10})t - 12(2^{-10})t^2 + 4t - 5t^2. \tag{15}$$

For each case of the model, the fitness functions are formulated as:

$$
\begin{aligned}
\in_{C-1} = {} & \frac{1}{N}\sum_{l=1}^{N}\left(t_m^2(2^{-4})\frac{d^2\hat{y}_m}{dt^2} + t_m\frac{d\hat{y}_m}{dt} + \hat{y}_m - 6(2^{-4})t_m^3 + 12(2^{-4})t_m^4 - 4t_m^3 + 5t_m^4\right)^2 \\
& + \frac{1}{2}\left((\hat{y}_0)^2 + (\hat{y}_N)^2\right),
\end{aligned}
$$

$$
\begin{aligned}
\in_{C-2} = {} & \frac{1}{N}\sum_{l=1}^{N}\left(t_m^2(2^{-6})\frac{d^2\hat{y}_m}{dt^2} + t_m\frac{d\hat{y}_m}{dt} + \hat{y}_m - 6(2^{-6})t_m^3 + 12(2^{-6})t_m^4 - 4t_m^3 + 5t_m^4\right)^2 \\
& + \frac{1}{2}\left((\hat{y}_0)^2 + (\hat{y}_N)^2\right),
\end{aligned}
$$

$$
\begin{aligned}
\in_{C-3} = {} & \frac{1}{N}\sum_{l=1}^{N}\left(t_m^2(2^{-8})\frac{d^2\hat{y}_m}{dt^2} + t_m\frac{d\hat{y}_m}{dt} + \hat{y}_m - 6(2^{-8})t_m^3 + 12(2^{-8})t_m^4 - 4t_m^3 + 5t_m^4\right)^2 \\
& + \frac{1}{2}\left((\hat{y}_0)^2 + (\hat{y}_N)^2\right),
\end{aligned}
$$

$$
\begin{aligned}
\in_{C-4} = {} & \frac{1}{N}\sum_{l=1}^{N}\left(t_m^2(2^{-10})\frac{d^2\hat{y}_m}{dt^2} + t_m\frac{d\hat{y}_m}{dt} + \hat{y}_m - 6(2^{-10})t_m^3 + 12(2^{-10})t_m^4 - 4t_m^3 + 5t_m^4\right)^2 \\
& + \frac{1}{2}\left((\hat{y}_0)^2 + (\hat{y}_N)^2\right).
\end{aligned}
$$

(16)

### Problem 2: Multi-singular singularly perturbed system

Consider the following SSP-BVP of second order involving trigonometric functions is given as [1]:

$$
\begin{aligned}
& \varepsilon\frac{d^2y}{dt^2} + \frac{1}{t}\frac{dy}{dt} + \frac{1}{t^2}y(t) = \sin\pi t(3 + \varepsilon(2 - t^2\pi^2)) + \pi t\cos\pi t(1 + 4\varepsilon), \\
& y(0) = 0, \ y(1) = 0.
\end{aligned}
$$

(17)

The exact solution of the above SSP-BVP (17) is provided as:

$$
y(t) = t^2 \sin \pi t.
$$

(18)

Four cases of the SSP-BVP (17) for taking small perturbation values, i.e., $\varepsilon = 2^{-4}, 2^{-6}, 2^{-8}$ and $2^{-10}$ Have been taken. The simplified form of the Eq (17) based on small parameters of $\varepsilon$ are written as:

$$
\begin{aligned}
2^{-4}\frac{d^2y}{dt^2} + \frac{1}{t}\frac{dy}{dt} + \frac{1}{t^2}y(t) &= \sin\pi t\left(3 + (2 - t^2\pi^2)2^{-4}\right) + \frac{5}{4}\pi t\cos\pi t, \\
2^{-6}\frac{d^2y}{dt^2} + \frac{1}{t}\frac{dy}{dt} + \frac{1}{t^2}y(t) &= \sin\pi t\left(3 + (2 - t^2\pi^2)2^{-6}\right) + \frac{17}{16}\pi t\cos\pi t, \\
2^{-8}\frac{d^2y}{dt^2} + \frac{1}{t}\frac{dy}{dt} + \frac{1}{t^2}y(t) &= \sin\pi t\left(3 + (2 - t^2\pi^2)2^{-8}\right) + \frac{65}{64}\pi t\cos\pi t, \\
2^{-10}\frac{d^2y}{dt^2} + \frac{1}{t}\frac{dy}{dt} + \frac{1}{t^2}y(t) &= \sin\pi t\left(3 + (2 - t^2\pi^2)2^{-10}\right) + \frac{257}{256}\pi t\cos\pi t.
\end{aligned}
$$

(19)

The fitness formulation of (17) becomes as:

$$
\begin{aligned}
\in_{C-1} &= \frac{1}{N}\sum_{l=1}^{N}\left(\begin{array}{c} \frac{1}{16}t_m^2\frac{d^2\hat{y}_m}{dt^2} + t_m\frac{d\hat{y}_m}{dt^2} + \hat{y}_m \\ -t_m^2\sin\pi t_m\left(3 + \frac{1}{16}\left(2 - t_m^2\pi^2\right)\right) - \frac{5}{4}\pi t_m^3\cos\pi t_m \end{array}\right)^2 + \frac{1}{2}\left((\hat{y}_0)^2 + (\hat{y}_N)^2\right), \\[2em]
\in_{C-2} &= \frac{1}{N}\sum_{l=1}^{N}\left(\begin{array}{c} \frac{1}{64}t_m^2\frac{d^2\hat{y}_m}{dt^2} + t_m\frac{d\hat{y}_m}{dt^2} + \hat{y}_m \\ -t_m^2\sin\pi t_m\left(3 + \frac{1}{64}\left(2 - t_m^2\pi^2\right)\right) - \frac{17}{16}\pi t_m^3\cos\pi t_m \end{array}\right)^2 + \frac{1}{2}\left((\hat{y}_0)^2 + (\hat{y}_N)^2\right), \\[2em]
\in_{C-3} &= \frac{1}{N}\sum_{l=1}^{N}\left(\begin{array}{c} \frac{1}{256}t_m^2\frac{d^2\hat{y}_m}{dt^2} + t_m\frac{d\hat{y}_m}{dt^2} + \hat{y}_m \\ -t_m^2\sin\pi t_m\left(3 + \frac{1}{256}\left(2 - t_m^2\pi^2\right)\right) - \frac{65}{64}\pi t_m^3\cos\pi t_m \end{array}\right)^2 + \frac{1}{2}\left((\hat{y}_0)^2 + (\hat{y}_N)^2\right), \\[2em]
\in_{C-4} &= \frac{1}{N}\sum_{l=1}^{N}\left(\begin{array}{c} \frac{1}{1024}t_m^2\frac{d^2\hat{y}_m}{dt^2} + t_m\frac{d\hat{y}_m}{dt^2} + \hat{y}_m - t_m^2\sin\pi t_m\left(3 + \frac{1}{1024}\left(2 - t_m^2\pi^2\right)\right) \\ -\frac{257}{256}\pi t_m^3\cos\pi t_m \end{array}\right)^2 + \frac{1}{2}\left((\hat{y}_0)^2 + (\hat{y}_N)^2\right).
\end{aligned}
\tag{20}
$$

To optimize the fitness functions (16) and (20), PSO-IPA is applied to achieve the approximate solutions for all four cases of SSP-BVP.

Figs 2 and 3 graphically represent the trained weights, the result's comparison for each case of problem 1 and problem 2 by taking 10 neurons. The approximate solution of both of the SSP-BVPs based on these weights. For results comparison, best, exact and mean results are plotted for both problems. It is clear that the best outcomes, mean solution and exact results matched to each other. Generally, for 100 independent runs best, mean and true results are same that proves the worth, exactness, stability and wider applicability of the proposed scheme. Figs 4 and 5 show the absolute error and performance indices for all cases of problem 1 and 2 supported by PSO-IPA. For absolute error (AE), best and worst results have been drawn in this regard. It is seen that for all cases of problem 1 and problem 2 the best values lie in the range of $10^{-05}$ to $10^{-07}$, whereas the worst measures are calculated $10^{-03}$ to $10^{-04}$ and $10^{-02}$ to $10^{-04}$. Performance measure based on the statistical values of the fitness, ENSE, TIC and MAD soundings have been plotted. In problem 1, the best fitness, TIC, MAD and ENSE values for each case lie around $10^{-10}$ to $10^{-12}$, $10^{-04}$ to $10^{-06}$, $10^{-08}$ to $10^{-10}$ and $10^{-06}$ to $10^{-08}$. The mean performances of these parameters lie around $10^{-04}$–$10^{-06}$, $10^{-04}$–$10^{-06}$, $10^{-08}$–$10^{-10}$ and $10^{-06}$–$10^{-08}$. The worst results of fitness lie between the ranges of $10^{-04}$ to $10^{-06}$, $10^{-02}$ to $10^{-04}$, $10^{-06}$ to $10^{-08}$ and $10^{-02}$ to $10^{-04}$. Hence, the worst measures of MAD, fitness, ENSE and TIC lie in good ranges. In problem 2, the best results of fitness, TIC, MAD and ENSE for all the cases lie around $10^{-10}$–$10^{-12}$, $10^{-04}$–$10^{-06}$, $10^{-08}$–$10^{-10}$ and $10^{-06}$–$10^{-08}$. Hence, the mean and worst fitness, MAD, TIC and ENSE performances also lie in the good measures.

Graphical designs of the fitness value are drawn in Figs 6 and 7 for all the cases of problem 1 and problem 2. The results indicates that more than 80% runs attained precise fitness values for all cases of both problems. The MAD values in convergence investigation for each case of both of the problems are graphically sketched in Figs 8 and 9. The achieved results are found to be in very good agreements. TIC graphical values of problem 1 and problem 2 are shown in Figs 10 and 11 shows that almost 85% runs accomplish reasonably accurate. The graphic

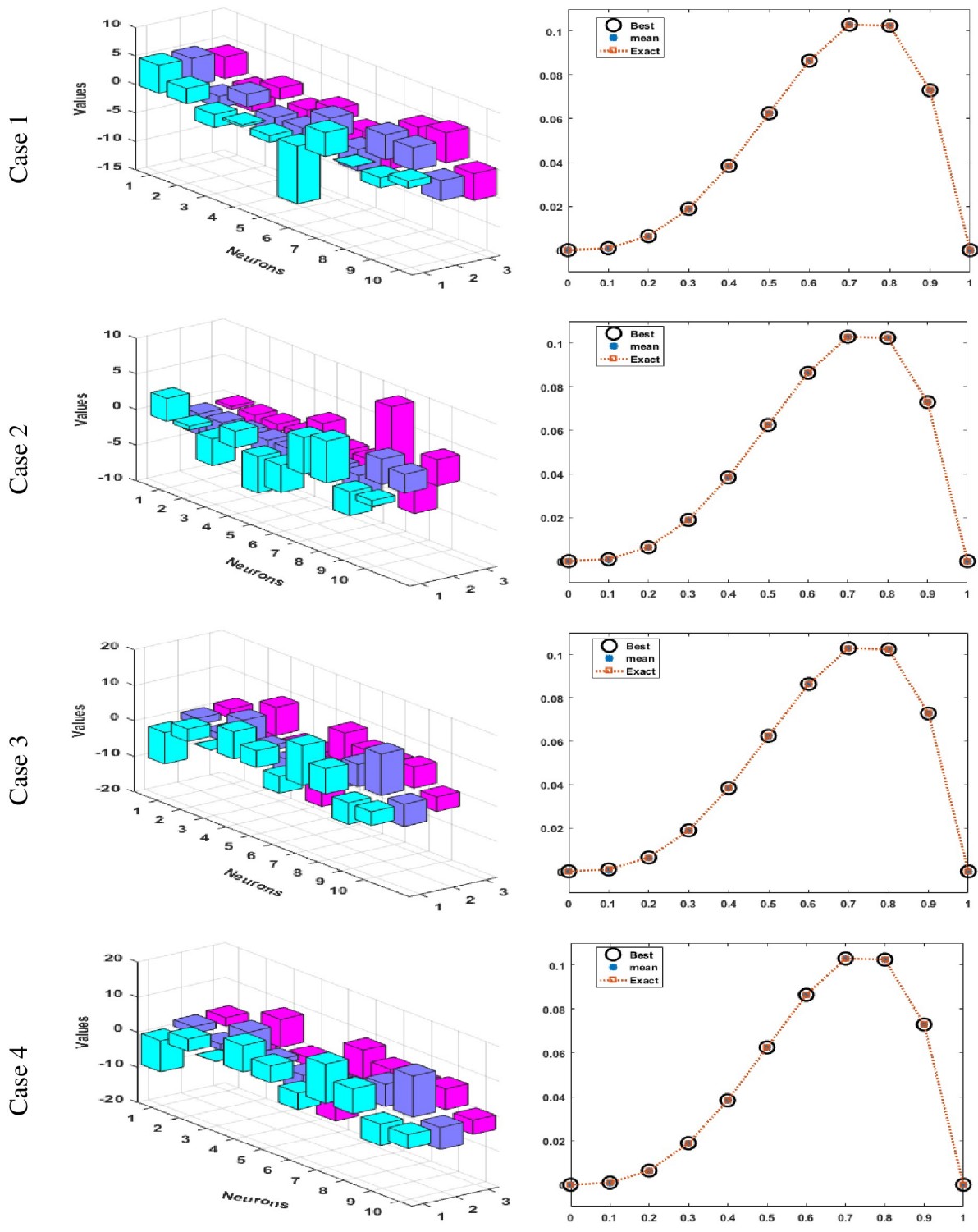

**Fig 2. Set of weights and comparison of result for all cases of problem 1.**

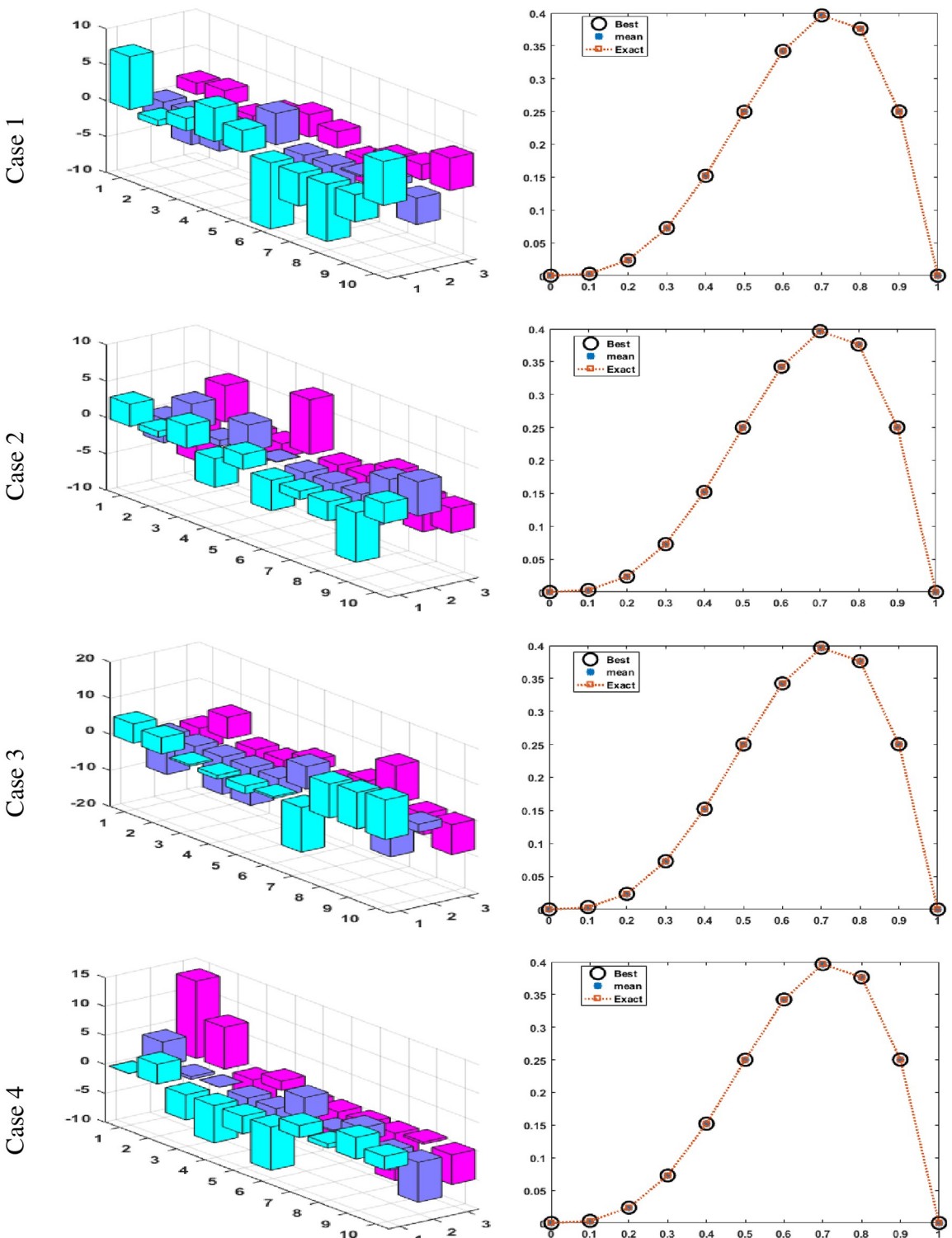

**Fig 3. Set of weights and comparison of result for all cases of problem 2.**

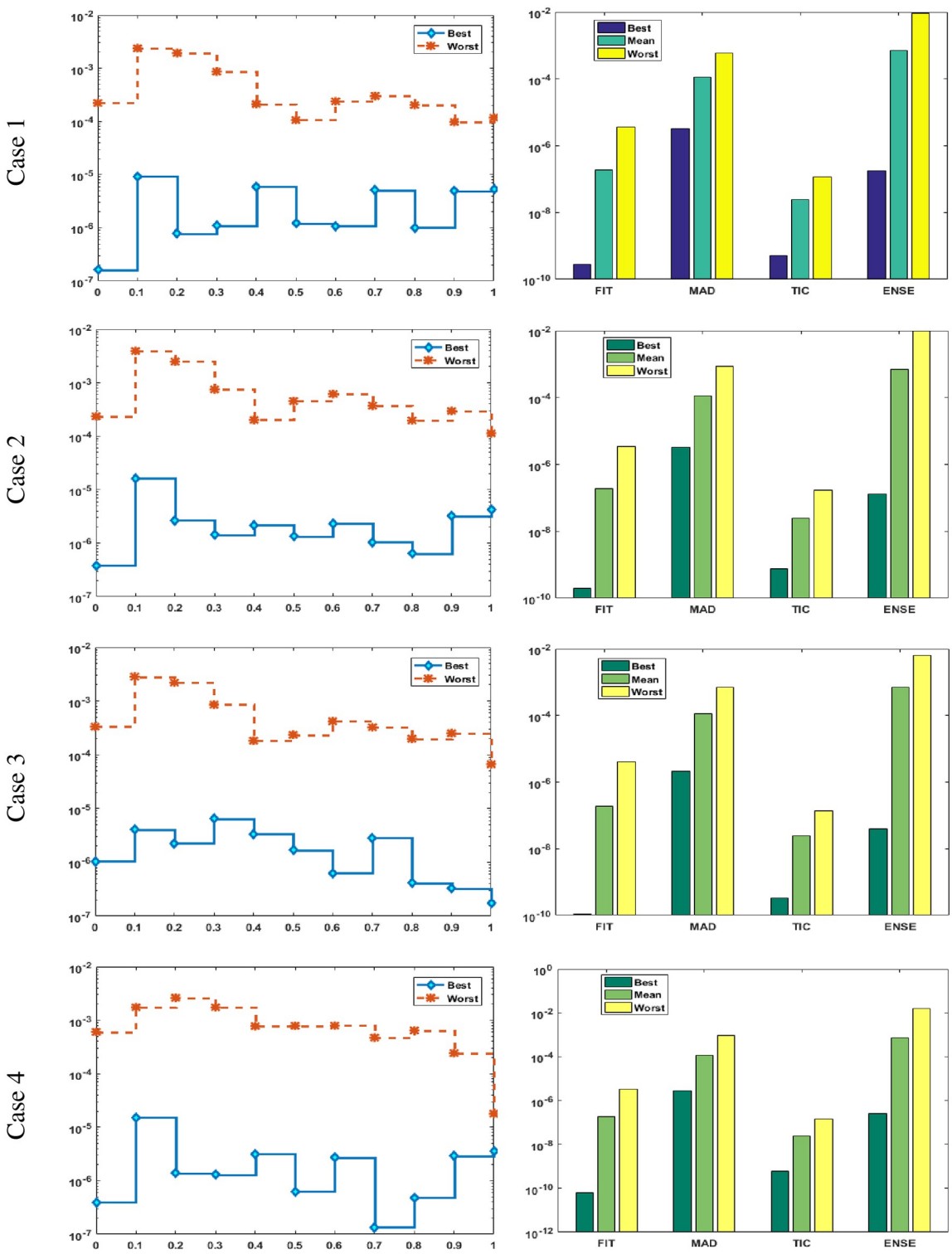

**Fig 4. Absolute error and performances for each case of problem 1.**

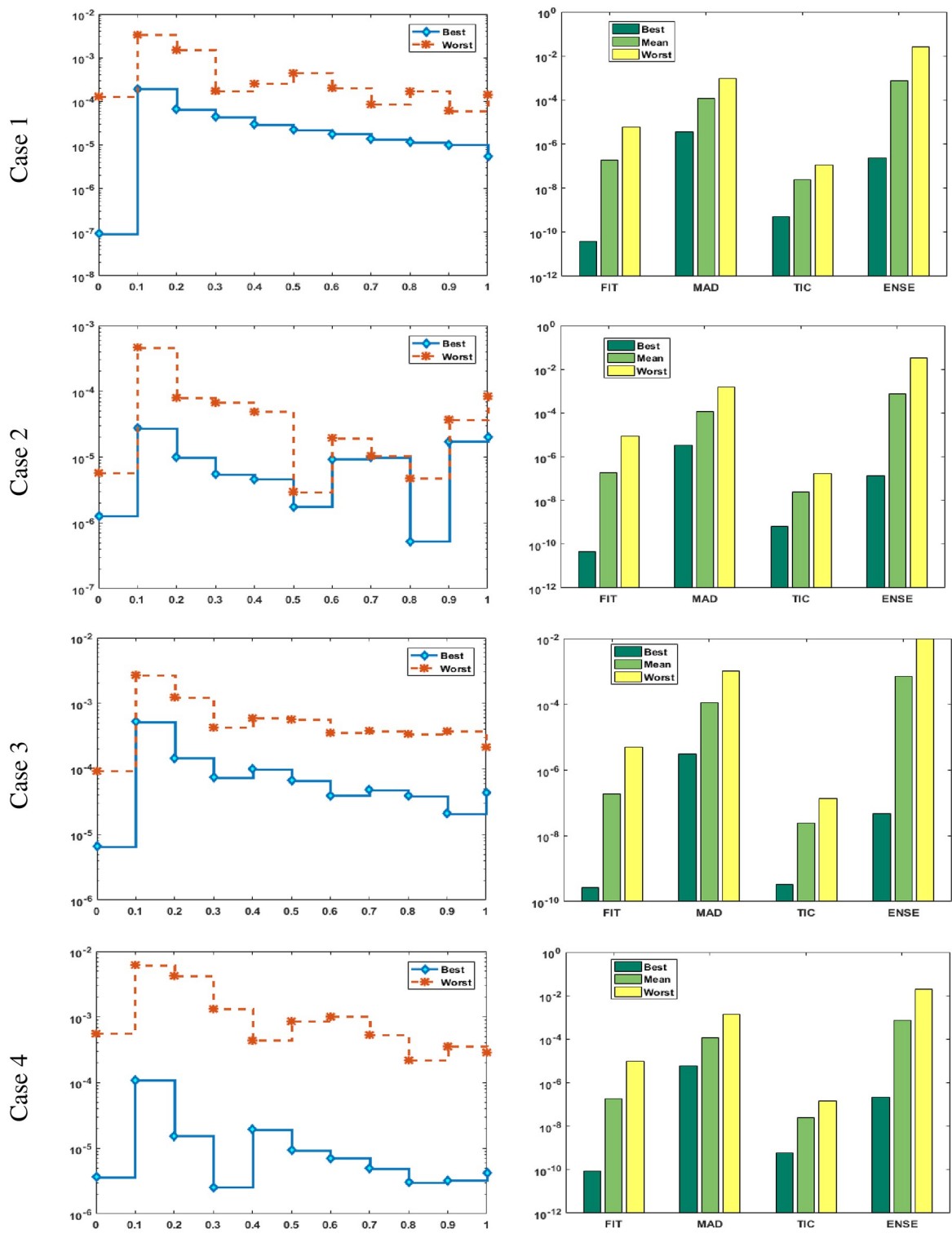

**Fig 5. Absolute error and performances for each case of problem 2.**

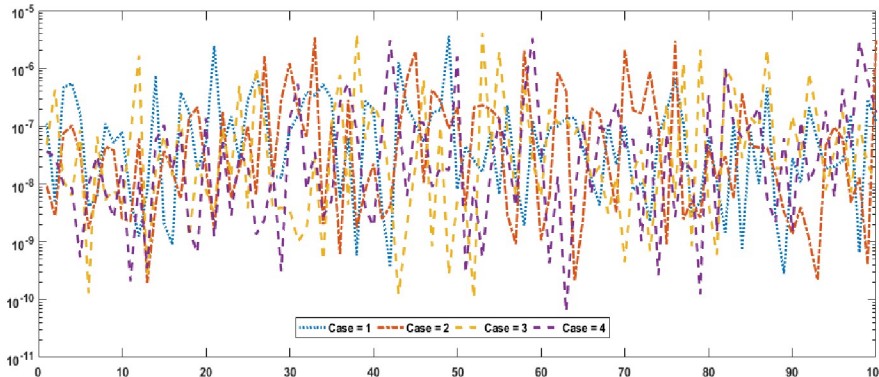

**Fig 6. Fitness values for each case of problem 1.**

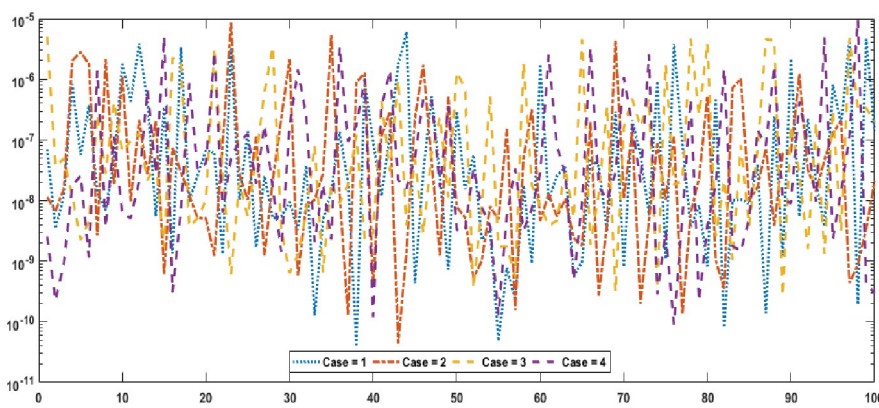

**Fig 7. Fitness values for each case of problem 2.**

standards of ENSE are represented in Figs 12 and 13. These results show that around 80% runs attain reasonably specific measures.

For more implications of the algorithm performance, the optimization has been made based on designed variables of ANN for 100 independent runs with PSO-IPA algorithm. The statistical term of minimum (MIN) values, mean values and standard deviation (STD)

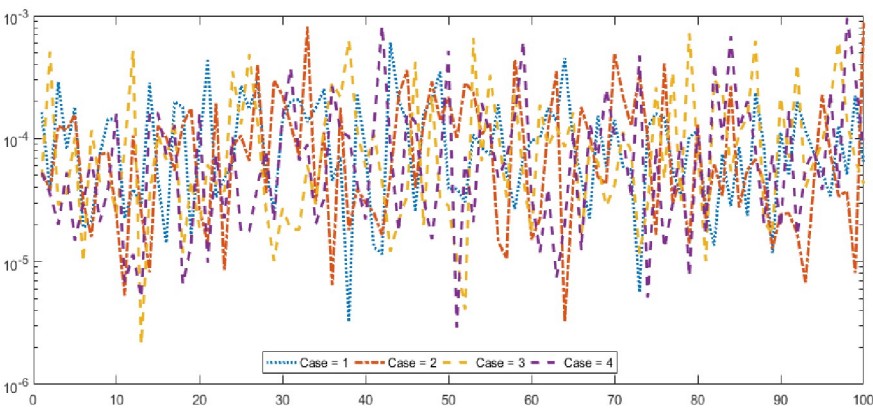

**Fig 8. MAD values for each case of problem 1.**

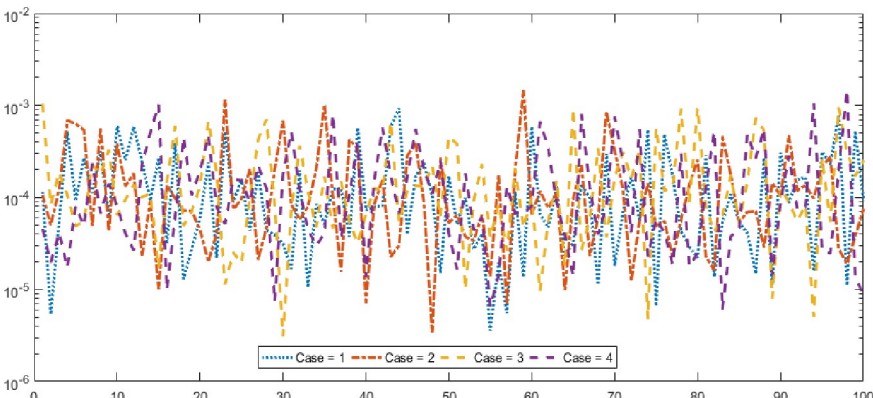

**Fig 9. MAD values for each case of problem 2.**

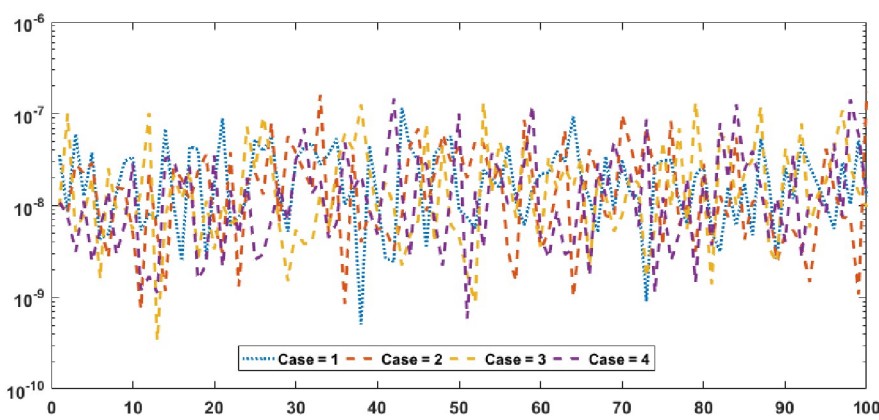

**Fig 10. TIC values for each case of problem 1.**

standards of AE are shown in Tables 3 and 4 for all the cases of problems (1–2). It is clear the achieved AE performances are calculated around $10^{-06}$ to $10^{-09}$ for MIN values, $10^{-04}$ to $10^{-05}$ for Mean and STD values. GFit, GMAD, GTIC and GENSE standards are calculated for all four cases of both of the problems are tabulated in Tables 5 and 6. The global performance

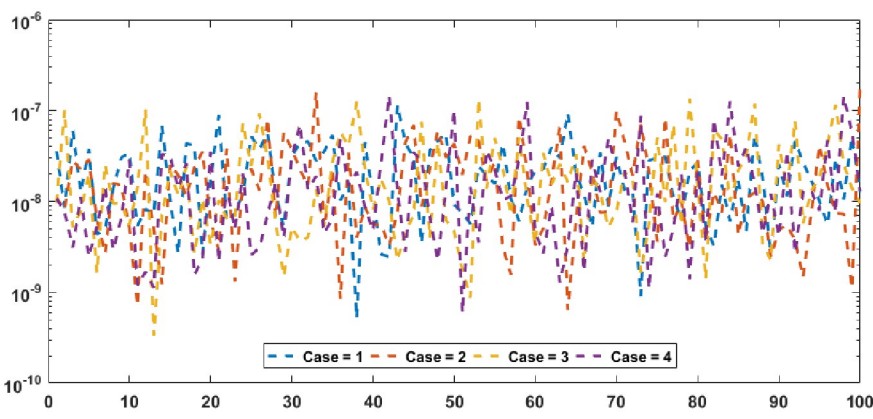

**Fig 11. TIC values for each case of problem 2.**

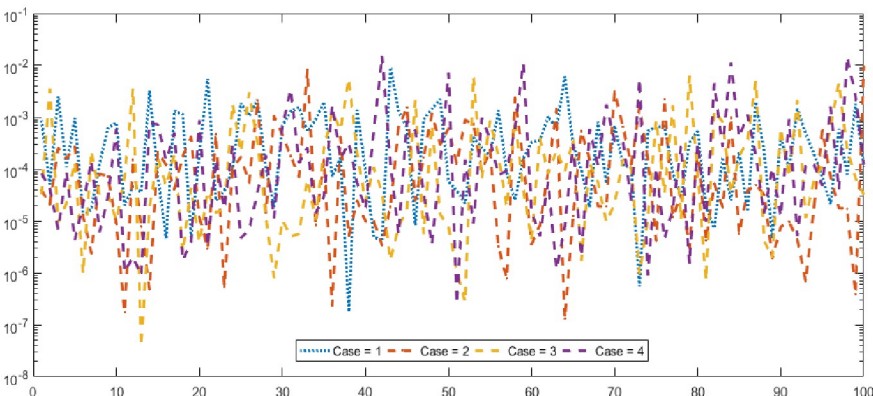

**Fig 12. ENSE values for each case of problem 1.**

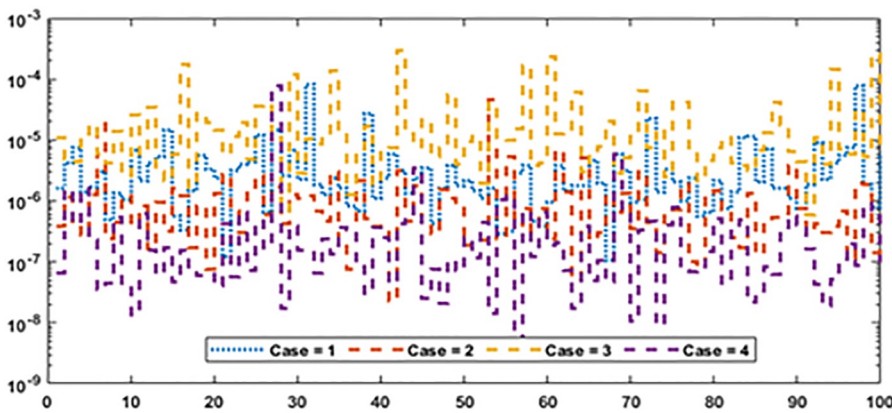

**Fig 13. ENSE values for each case of problem 2.**

**Table 3. Statistics results of AE for each case of problem 1.**

| $t$ | $\varepsilon = 2^{-4}$ | | | $\varepsilon = 2^{-6}$ | | | $\varepsilon = 2^{-8}$ | | | $\varepsilon = 2^{-10}$ | | |
|---|---|---|---|---|---|---|---|---|---|---|---|---|
| | Min | Mean | STD | Min | Mean | STD | Min | Mean | STD | Min | Mean | STD |
| 0 | $5 \times 10^{-08}$ | $2 \times 10^{-05}$ | $3 \times 10^{-05}$ | $2 \times 10^{-08}$ | $3 \times 10^{-05}$ | $5 \times 10^{-05}$ | $1 \times 10^{-07}$ | $3 \times 10^{-05}$ | $6 \times 10^{-05}$ | $1 \times 10^{-08}$ | $3 \times 10^{-05}$ | $7 \times 10^{-05}$ |
| 0.1 | $4 \times 10^{-06}$ | $5 \times 10^{-04}$ | $5 \times 10^{-04}$ | $7 \times 10^{-07}$ | $5 \times 10^{-04}$ | $6 \times 10^{-04}$ | $4 \times 10^{-06}$ | $6 \times 10^{-04}$ | $7 \times 10^{-04}$ | $2 \times 10^{-06}$ | $5 \times 10^{-04}$ | $6 \times 10^{-04}$ |
| 0.2 | $7 \times 10^{-07}$ | $2 \times 10^{-04}$ | $2 \times 10^{-04}$ | $1 \times 10^{-06}$ | $2 \times 10^{-04}$ | $4 \times 10^{-04}$ | $2 \times 10^{-06}$ | $3 \times 10^{-04}$ | $4 \times 10^{-04}$ | $5 \times 10^{-07}$ | $2 \times 10^{-04}$ | $4 \times 10^{-04}$ |
| 0.3 | $5 \times 10^{-07}$ | $7 \times 10^{-05}$ | $1 \times 10^{-04}$ | $6 \times 10^{-08}$ | $8 \times 10^{-05}$ | $1 \times 10^{-04}$ | $6 \times 10^{-07}$ | $1 \times 10^{-04}$ | $2 \times 10^{-04}$ | $3 \times 10^{-07}$ | $1 \times 10^{-04}$ | $2 \times 10^{-04}$ |
| 0.4 | $1 \times 10^{-07}$ | $6 \times 10^{-05}$ | $8 \times 10^{-05}$ | $1 \times 10^{-06}$ | $6 \times 10^{-05}$ | $6 \times 10^{-05}$ | $3 \times 10^{-06}$ | $7 \times 10^{-05}$ | $1 \times 10^{-04}$ | $3 \times 10^{-06}$ | $7 \times 10^{-05}$ | $1 \times 10^{-04}$ |
| 0.5 | $1 \times 10^{-06}$ | $7 \times 10^{-05}$ | $6 \times 10^{-05}$ | $1 \times 10^{-06}$ | $6 \times 10^{-05}$ | $7 \times 10^{-05}$ | $1 \times 10^{-06}$ | $7 \times 10^{-05}$ | $8 \times 10^{-05}$ | $6 \times 10^{-07}$ | $7 \times 10^{-05}$ | $1 \times 10^{-04}$ |
| 0.6 | $4 \times 10^{-09}$ | $6 \times 10^{-05}$ | $6 \times 10^{-05}$ | $2 \times 10^{-07}$ | $7 \times 10^{-05}$ | $9 \times 10^{-05}$ | $3 \times 10^{-07}$ | $7 \times 10^{-05}$ | $1 \times 10^{-04}$ | $5 \times 10^{-07}$ | $6 \times 10^{-05}$ | $1 \times 10^{-04}$ |
| 0.7 | $1 \times 10^{-07}$ | $3 \times 10^{-05}$ | $5 \times 10^{-05}$ | $1 \times 10^{-07}$ | $5 \times 10^{-05}$ | $8 \times 10^{-05}$ | $2 \times 10^{-07}$ | $5 \times 10^{-05}$ | $9 \times 10^{-05}$ | $5 \times 10-09$ | $5 \times 10^{-05}$ | $9 \times 10^{-05}$ |
| 0.8 | $1 \times 10^{-06}$ | $2 \times 10^{-05}$ | $3 \times 10^{-05}$ | $1 \times 10^{-07}$ | $3 \times 10^{-05}$ | $3 \times 10^{-05}$ | $1 \times 10-08$ | $3 \times 10^{-05}$ | $5 \times 10^{-05}$ | $4 \times 10^{-07}$ | $3 \times 10^{-05}$ | $7 \times 10^{-05}$ |
| 0.9 | $6 \times 10^{-07}$ | $3 \times 10^{-05}$ | $3 \times 10^{-05}$ | $2 \times 10^{-07}$ | $4 \times 10^{-05}$ | $4 \times 10^{-05}$ | $3 \times 10^{-07}$ | $4 \times 10^{-05}$ | $5 \times 10^{-05}$ | $1 \times 10^{-07}$ | $4 \times 10^{-05}$ | $5 \times 10^{-05}$ |
| 1.0 | $3 \times 10^{-07}$ | $2 \times 10^{-05}$ | $2 \times 10^{-05}$ | $2 \times 10^{-07}$ | $3 \times 10^{-05}$ | $5 \times 10^{-05}$ | $1 \times 10^{-07}$ | $4 \times 10^{-05}$ | $6 \times 10^{-05}$ | $3 \times 10^{-07}$ | $4 \times 10^{-05}$ | $6 \times 10^{-05}$ |

**Table 4. Statistics results of AE for each case of problem 2.**

| $t$ | $\varepsilon = 2^{-4}$ | | | $\varepsilon = 2^{-6}$ | | | $\varepsilon = 2^{-8}$ | | | $\varepsilon = 2^{-10}$ | | |
|---|---|---|---|---|---|---|---|---|---|---|---|---|
| | Min | Mean | STD | Min | Mean | STD | Min | Mean | STD | Min | Mean | STD |
| 0 | $1\times10^{-08}$ | $2\times10^{-05}$ | $4\times10^{-05}$ | $6\times10^{-08}$ | $3\times10^{-05}$ | $6\times10^{-05}$ | $7\times10{-09}$ | $3\times10^{-05}$ | $7\times10^{-05}$ | $2\times10^{-08}$ | $3\times10^{-05}$ | $7\times10^{-05}$ |
| 0.1 | $3\times10^{-06}$ | $8\times10^{-04}$ | $1\times10^{-03}$ | $5\times10^{-06}$ | $8\times10^{-04}$ | $1\times10^{-03}$ | $4\times10^{-06}$ | $9\times10^{-04}$ | $1\times10^{-03}$ | $1\times10^{-06}$ | $9\times10{-04}$ | $1\times10^{-03}$ |
| 0.2 | $6\times10^{-07}$ | $3\times10^{-04}$ | $5\times10^{-04}$ | $1\times10^{-06}$ | $3\times10^{-04}$ | $6\times10^{-04}$ | $2\times10^{-06}$ | $4\times10^{-04}$ | $6\times10^{-04}$ | $3\times10^{-06}$ | $4\times10^{-04}$ | $6\times10^{-04}$ |
| 0.3 | $4\times10^{-07}$ | $8\times10^{-05}$ | $1\times10^{-04}$ | $4\times10^{-06}$ | $1\times10^{-04}$ | $2\times10^{-04}$ | $8\times10^{-07}$ | $1\times10^{-04}$ | $1\times10^{-04}$ | $2\times10^{-06}$ | $1\times10^{-04}$ | $1\times10^{-04}$ |
| 0.4 | $3\times10^{-07}$ | $9\times10^{-05}$ | $9\times10^{-05}$ | $1\times10^{-06}$ | $1\times10^{-04}$ | $1\times10^{-04}$ | $2\times10^{-06}$ | $1\times10^{-04}$ | $1\times10^{-04}$ | $2\times10^{-07}$ | $1\times10^{-04}$ | $1\times10^{-04}$ |
| 0.5 | $9\times10^{-07}$ | $1\times10^{-04}$ | $1\times10^{-04}$ | $1\times10^{-06}$ | $1\times10^{-04}$ | $1\times10^{-04}$ | $1\times10^{-07}$ | $1\times10^{-04}$ | $2\times10^{-04}$ | $7\times10{-09}$ | $1\times10^{-04}$ | $2\times10^{-04}$ |
| 0.6 | $5\times10^{-07}$ | $8\times10^{-05}$ | $1\times10^{-04}$ | $2\times10^{-06}$ | $8\times10^{-05}$ | $1\times10^{-04}$ | $2\times10^{-06}$ | $9\times10^{-05}$ | $1\times10^{-04}$ | $1\times10^{-07}$ | $1\times10^{-04}$ | $1\times10^{-04}$ |
| 0.7 | $1\times10^{-07}$ | $4\times10^{-05}$ | $6\times10^{-05}$ | $7\times10^{-08}$ | $5\times10^{-05}$ | $9\times10^{-05}$ | $6\times10^{-07}$ | $6\times10^{-05}$ | $6\times10^{-05}$ | $1\times10^{-06}$ | $6\times10^{-05}$ | $9\times10^{-05}$ |
| 0.8 | $1\times10^{-07}$ | $3\times10^{-05}$ | $4\times10^{-05}$ | $2\times10^{-07}$ | $5\times10^{-05}$ | $8\times10^{-05}$ | $3\times10^{-07}$ | $7\times10^{-05}$ | $9\times10^{-05}$ | $5\times10^{-08}$ | $6\times10^{-05}$ | $8\times10^{-05}$ |
| 0.9 | $1\times10^{-07}$ | $5\times10^{-05}$ | $7\times10^{-05}$ | $1\times10^{-07}$ | $5\times10^{-05}$ | $9\times10^{-05}$ | $2\times10^{-07}$ | $7\times10^{-05}$ | $1\times10^{-04}$ | $2\times10^{-07}$ | $7\times10^{-05}$ | $1\times10^{-04}$ |
| 1.0 | $2\times10^{-07}$ | $3\times10^{-05}$ | $3\times10^{-05}$ | $3\times10^{-07}$ | $5\times10^{-05}$ | $7\times10^{-05}$ | $7\times10^{-08}$ | $5\times10^{-05}$ | $7\times10^{-05}$ | $2\times10^{-07}$ | $6\times10^{-05}$ | $8\times10^{-05}$ |

**Table 5. Global performances for problem 1.**

| Case | GFIT | | GMAD | | GTIC | | GENSE | |
|---|---|---|---|---|---|---|---|---|
| | Mag | STD | Mag | STD | Mag | STD | Mag | STD |
| 1 | 1.89E-07 | 4.61E-07 | 1.15E-04 | 1.04E-04 | 2.43E-08 | 2.14E-08 | 7.17E-04 | 1.32E-03 |
| 2 | 2.69E-07 | 6.51E-07 | 1.21E-04 | 1.51E-04 | 2.38E-08 | 2.98E-08 | 4.65E-04 | 1.33E-03 |
| 3 | 2.72E-07 | 6.94E-07 | 1.34E-04 | 1.66E-04 | 2.62E-08 | 3.22E-08 | 6.02E-04 | 1.35E-03 |
| 4 | 1.87E-07 | 5.56E-07 | 1.17E-04 | 1.68E-04 | 2.20E-08 | 2.99E-08 | 9.78E-04 | 2.74E-03 |

**Table 6. Global performances for problem 2.**

| Case | GFIT | | GMAD | | GTIC | | GENSE | |
|---|---|---|---|---|---|---|---|---|
| | Mag | STD | Mag | STD | Mag | STD | Mag | STD |
| 1 | 4.61E-07 | 1.13E-06 | 1.62E-04 | 1.91E-04 | 2.42E-08 | 2.14E-08 | 1.97E-03 | 4.21E-03 |
| 2 | 4.17E-07 | 1.18E-06 | 1.79E-04 | 2.52E-04 | 2.38E-08 | 2.99E-08 | 1.66E-03 | 4.62E-03 |
| 3 | 5.66E-07 | 1.26E-06 | 2.00E-04 | 2.40E-04 | 2.61E-08 | 3.23E-08 | 9.66E-04 | 2.13E-03 |
| 4 | 5.06E-07 | 1.33E-06 | 1.97E-04 | 2.62E-04 | 2.20E-08 | 2.99E-08 | 1.01E-03 | 2.61E-03 |

operators achieve generally for all cases of problems (1–2) as a result. Complexity study (CS) of proposed methodology is presented based on average optimizer value, i.e., execution time CS-ET, averagely number of iterations performed by the technique, i.e., CS-NG and averagely evaluated function. These measures are determined of CS operators for 100 trials of the procedure and outcomes are shown in Tables 7 and 8 together with the values of Mean and STD for all four cases of problems (1–2).

**Table 7. Complexity representations for problem 1.**

| Case | Time of implementation | | Iteration | | Function Counts | |
|---|---|---|---|---|---|---|
| | Mean | STD | Mean | Mean | STD | Mean |
| 1 | 48.577183 | 17.960816 | 5172.62 | 2095.5142 | 270919.98 | 104853.54 |
| 2 | 50.484556 | 16.202738 | 5440.34 | 1931.4327 | 284239.2 | 96485.599 |
| 3 | 50.544832 | 17.660773 | 5418.86 | 2081.4746 | 283190.26 | 104019.6 |
| 4 | 52.240054 | 17.516587 | 5525.54 | 2026.1252 | 288466.43 | 101466.06 |

**Table 8. Complexity representations for problem 2.**

| Case | Time of implementation | | Iteration | | Function Counts | |
|------|------|------|------|------|------|------|
| | Mean | STD | Mean | Mean | STD | Mean |
| 1 | 52.898577 | 14.941427 | 5771.11 | 1783.268 | 300885.18 | 89322.19 |
| 2 | 53.606204 | 15.638092 | 5835.87 | 1844.6481 | 304084.78 | 92248.666 |
| 3 | 52.216913 | 16.385791 | 5586.94 | 1935.3087 | 291785.94 | 96741.717 |
| 4 | 56.627975 | 12.204644 | 6218.05 | 1437.3541 | 323169.83 | 71906.603 |

## Conclusions

The present work is summarized as:

- A computational intelligent charter is settled efficiently for two problems along with four cases of SSP-BVPs by exploiting the universal approximation strength of feed-forward ANNs optimized with PSO supported with IPA.

- The proposed and exact solutions matched for all cases of both examples, which indicates the exactness and worth of the methodology in convergence and accuracy sense.

- Min, Mean and STD statistical operators indicate the proposed scheme give reliably these operators value very close to zero in each case of both of the problems.

- The statistical performance of fitness, TIC, MAD and ENSE and their global form is used to check the accurateness and convergence of the scheme.

- The complexity performances of the proposed ANNs-PSO-IPA for solving the SSP-BVPs using the operators through the values of time, function counts and generations during the practice to check the optimization of the design network variables, which indicate the smooth accomplishment of the SSP-BVPs.

## Author Contributions

**Conceptualization:** Zulqurnain Sabir, Thongchai Botmart.

**Data curation:** Thongchai Botmart, Muhammad Asif Zahoor Raja, Wajaree Weera, Fevzi Erdoğan.

**Formal analysis:** Thongchai Botmart, Wajaree Weera, Fevzi Erdoğan.

**Funding acquisition:** Zulqurnain Sabir, Thongchai Botmart.

**Investigation:** Zulqurnain Sabir.

**Methodology:** Zulqurnain Sabir, Thongchai Botmart, Wajaree Weera, Fevzi Erdoğan.

**Project administration:** Fevzi Erdoğan.

**Software:** Muhammad Asif Zahoor Raja, Wajaree Weera.

**Supervision:** Muhammad Asif Zahoor Raja.

**Validation:** Muhammad Asif Zahoor Raja.

**Writing – original draft:** Thongchai Botmart.

**Writing – review & editing:** Thongchai Botmart, Wajaree Weera.

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
