## [Decision Letter · Decision Letter 0]

14 Mar 2022

PONE-D-21-38900A stochastic numerical approach for a class of singular singularly perturbed systemPLOS ONE

Dear Dr. Botmart,

Thank you for submitting your manuscript to PLOS ONE. After careful consideration, we feel that it has merit but does not fully meet PLOS ONE’s publication criteria as it currently stands. Therefore, we invite you to submit a revised version of the manuscript that addresses the points raised during the review process.

We look forward to receiving your revised manuscript.

Kind regards,

Chengming Huang

Academic Editor

PLOS ONE

Journal Requirements:

4. We note you have included a table to which you do not refer in the text of your manuscript. Please ensure that you refer to table 2 in your text; if accepted, production will need this reference to link the reader to the Table.

Reviewers' comments:

Reviewer's Responses to Questions

**Comments to the Author**

1. Is the manuscript technically sound, and do the data support the conclusions?

Reviewer #1: Yes

Reviewer #2: Yes

2. Has the statistical analysis been performed appropriately and rigorously? 

Reviewer #1: Yes

Reviewer #2: Yes

3. Have the authors made all data underlying the findings in their manuscript fully available?

Reviewer #1: Yes

Reviewer #2: Yes

4. Is the manuscript presented in an intelligible fashion and written in standard English?

Reviewer #1: Yes

Reviewer #2: Yes

5. Review Comments to the Author

Reviewer #1: A stochastic numerical approach for a class of singular singularly perturbed system

In the present study, a neuro-evolutionary scheme is presented for solving a class of singular singularly perturbed boundary value problems (SSP-BVPs) by manipulating the strength of feed-forward artificial neural networks (ANNs), global search particle swarm optimization (PSO), and local search interior-point algorithm (IPA), i.e., ANNs-PSO-IPA. An error-based fitness function is designed using the differential form of the SSP-BVPs and its boundary conditions. The optimization of this fitness function is performed by using the computing capabilities of ANNs-PSO-IPA. Four cases of two SSP systems are tested to confirm the performance of the suggested ANNs-PSO-IPA. The correctness of the scheme is observed by using the comparison of the proposed and the exact solutions. The performance indices through different statistical operators are also provided to solve the SSP-BVPs using the proposed ANNs-PSO-IPA. Moreover, the reliability of the scheme is observed by taking hundred independent executions, and different statistical performances have been provided for solving the SSP-BVPs to check the convergence, robustness, and accuracy.

As we all know, the models based on the singular systems are advantageous, and the solution of such models by using stochastic schemes is also appreciable. The ANNs methods are crucial and have considerable applications in diverse fields

After carefully reviewing the paper, I conclude that:

• Paper is interesting

• Results are mathematically correct

• Presentation is also nice

However, the paper can be accepted for publication after considering the following minor revisions as follows:

1)The introduction is written well, but add a few more recent applications for improvement, for instance:

https://doi.org/10.1016/j.asej.2020.11.007

https://doi.org/10.1371/journal.pone.0235829

10.1109/ACCESS.2021.3078750

https://doi.org/10.1155/2022/9660746

10.1109/ACCESS.2021.3133815

2) Adjust statistical performance measures in section 2

3) Use proper punctuation

4) Use the same style of references

5) What effects can be seen by increasing or decreasing the neurons

6) Mention a few applications of IPA.

Reviewer #2: In this paper, a method based on numerical approach is developed for solving singularly perturbed system.

This paper will be recommended for publication after some revisions.

Comments:

1. Complexity: Please give some theoretical analysis in the time complexity of the current paper.

2. Authors improve the grammar in the manuscript.

3. Authors should add error analysis in details of proposed method.

4. To verify the convergence and efficiency of the method it would be very important to show how the error changes as I increase. I think that such tables should be presented.

5. I think that the numerical results presented are not sufficient to convince the reader

about the advantage of the method.

6. The list of references is quite poor and it should be enlarged.

6. PLOS authors have the option to publish the peer review history of their article (what does this mean?). If published, this will include your full peer review and any attached files.

Reviewer #1: No

Reviewer #2: No

---

## [Author Response · Author response to Decision Letter 0]

21 Apr 2022

Author Response

Paper Title: 

A stochastic numerical approach for a class of singular singularly perturbed system 

By: Zulqurnain Sabir, Thongchai Botmart, Muhammad Asif Zahoor Raja, Wajaree weera, Fevzi Erdoğan

ID: PONE-D-21-38900, Journal: PLOS ONE

Reviewers' comments:

Reviewer 1: General comments

In the present study, a neuro-evolutionary scheme is presented for solving a class of singular singularly perturbed boundary value problems (SSP-BVPs) by manipulating the strength of feed-forward artificial neural networks (ANNs), global search particle swarm optimization (PSO), and local search interior-point algorithm (IPA), i.e., ANNs-PSO-IPA. An error-based fitness function is designed using the differential form of the SSP-BVPs and its boundary conditions. The optimization of this fitness function is performed by using the computing capabilities of ANNs-PSO-IPA. Four cases of two SSP systems are tested to confirm the performance of the suggested ANNs-PSO-IPA. The correctness of the scheme is observed by using the comparison of the proposed and the exact solutions. The performance indices through different statistical operators are also provided to solve the SSP-BVPs using the proposed ANNs-PSO-IPA. Moreover, the reliability of the scheme is observed by taking hundred independent executions, and different statistical performances have been provided for solving the SSP-BVPs to check the convergence, robustness, and accuracy.

As we all know, the models based on the singular systems are advantageous, and the solution of such models by using stochastic schemes is also appreciable. The ANNs methods are crucial and have considerable applications in diverse fields

After carefully reviewing the paper, I conclude that:

• Paper is interesting

• Results are mathematically correct

• Presentation is also nice

However, the paper can be accepted for publication after considering the following minor revisions as follows:

Authors reply

First of all, many thanks for your valuable remarks “As we all know, the models based on the singular systems are advantageous, and the solution of such models by using stochastic schemes is also appreciable. The ANNs methods are crucial and have considerable applications in diverse fields

After carefully reviewing the paper, I conclude that:

• Paper is interesting

• Results are mathematically correct

• Presentation is also nice” 

 and favorable recommendation “the paper can be accepted for publication after considering the following minor revisions” on our submitted manuscript. Moreover, the authors tried level best to address all the valuable suggestions of the worthy anonymous reviewer and modified the manuscript accordingly.

Query 1

1) The introduction is written well, but add a few more recent applications for improvement, for instance:

https://doi.org/10.1016/j.asej.2020.11.007

https://doi.org/10.1371/journal.pone.0235829

10.1109/ACCESS.2021.3078750

https://doi.org/10.1155/2022/9660746

10.1109/ACCESS.2021.3133815

Authors Reply

Agreed. We have updated the manuscript by providing the exhaustive literature review in the introduction section to portray the problem statement with justification with the help of relevant, recent and reputed journal articles as suggested. Many thanks for suggesting the relevant recent articles [r1-r8] on the topic and indeed these article are very helpful to improve the technical and presentation strength of the introduction section. 

[r1] Ahmad, A., Sulaiman, M., Aljohani, A.J., Alhindi, A. and Alrabaiah, H., 2021. Design of an efficient algorithm for solution of Bratu differential equations. Ain Shams Engineering Journal, 12(2), pp.2211-2225.

[r2] Waseem, W., Sulaiman, M., Kumam, P., Shoaib, M., Raja, M.A.Z. and Islam, S., 2020. Investigation of singular ordinary differential equations by a neuroevolutionary approach. Plos one, 15(7), p.e0235829.

[r3] Zhang, Y., Lin, J., Hu, Z., Khan, N.A. and Sulaiman, M., 2021. Analysis of third-order nonlinear multi-singular Emden–Fowler equation by using the LeNN-WOA-NM algorithm. IEEE Access, 9, pp.72111-72138.

[r4] Fawad Khan, M., Bonyah, E., Alshammari, F.S., Ghufran, S.M. and Sulaiman, M., 2022. Modelling and Analysis of Virotherapy of Cancer Using an Efficient Hybrid Soft Computing Procedure. Complexity, 2022.

[r5] Rahman, I.U., Sulaiman, M., Alarfaj, F.K., Kumam, P. and Laouini, G., 2021. Investigation of Non-Linear MHD Jeffery–Hamel Blood Flow Model Using a Hybrid Metaheuristic Approach. IEEE Access, 9, pp.163214-163232.

[r6] Bukhari, A.H., Raja, M.A.Z., Sulaiman, M., Islam, S., Shoaib, M. and Kumam, P., 2020. Fractional neuro-sequential ARFIMA-LSTM for financial market forecasting. IEEE Access, 8, pp.71326-71338.

[r7] Bukhari, A.H., Sulaiman, M., Raja, M.A.Z., Islam, S., Shoaib, M. and Kumam, P., 2020. Design of a hybrid NAR-RBFs neural network for nonlinear dusty plasma system. Alexandria Engineering Journal, 59(5), pp.3325-3345.

[r8] Bukhari, A.H., Sulaiman, M., Islam, S., Shoaib, M., Kumam, P. and Raja, M.A.Z., 2020. Neuro-fuzzy modeling and prediction of summer precipitation with application to different meteorological stations. Alexandria Engineering Journal, 59(1), pp.101-116.

Query 2

2) Adjust statistical performance measures in section 2

Authors reply

Agreed. We have updated the manuscript by the adjusting the statistical performance measures in section 2 as suggested.

Query 3

3) Use proper punctuation

Authors reply

Agreed. We have updated the manuscript by careful review of the inserted punctuations in the whole draft as suggested.

Query 4

4) Use the same style of references

Authors reply

Agreed. We have updated the manuscript by all the references on consistent template as suggested.

Query 5

5) What effect can be seen by increasing or decreasing the neurons

Authors reply

By increasing the number of neurons, the a slight improvement in the accuracy is achieved but at the cost of additional computations and also arises an issue of overfitting.

Query 6

6) Mention a few applications of IPA.

Authors reply

Agreed. We have updated the manuscript by mentioning the applications of the IPA as suggested.

Reviewer 2 General Comments

Reviewer #2: In this paper, a method based on numerical approach is developed for solving singularly perturbed system.

This paper will be recommended for publication after some revisions.

Authors reply

First of all, many thanks for your valuable remarks “a method based on numerical approach is developed for solving singularly perturbed system” and favorable recommendations “recommended for publication after some revisions” on our submitted manuscript. Additionally, the authors tried their level best to address all the changes suggested by the worthy anonymous reviewer and the manuscript is modified accordingly.

Comment 1 

1. Complexity: Please give some theoretical analysis in the time complexity of the current paper.

Authors reply

Agreed. We have updated the manuscript by providing the necessary information regarding the complexity in the results section of the revised manuscript.

Please see the complexity information presented in highlighted Tables 7 and 8 along with the interpretations in the results section of the revised manuscript.

Comment 2 

2. Authors improve the grammar in the manuscript.

Authors reply

Agreed. We have updated the manuscript by critical, careful and exhaustive review of the whole draft to improve the linguistic quality by avoiding the grammatical errors, ambiguous sentences, and typos as suggested by the worthy anonymous reviewer for better understanding of the readers.

Comment 3 

3. Authors should add error analysis in details of proposed method.

Authors reply

Agreed. We have updated the manuscript by providing the rigorous error analysis from the available reference solutions for the both problems in terms of difference performance indices in order to portray the inferences and worth of the design methodology as suggested for better understanding of readers.

Comment 4 

4. To verify the convergence and efficiency of the method it would be very important to show how the error changes as I increase. I think that such tables should be presented.

Authors reply

Agreed. We have updated the manuscript by providing information regarding the convergence and efficient more elaborative on the basis on single and multiple runs of the presented scheme as suggested.

In this study, the accuracy, convergence, stability and robustness of the presented scheme is endorsed with the help of rigorous statistics via performance measure on the basis of mean absolute deviation (MAD), Theil’s inequality coefficient (TIC) operator and Nash Sutcliffe efficiency (NSE) operator as well as global MAD, Global TIC and Global NSE are applied to solve SSP-BVPs for all cases of both problems. Please see the results section of the revised manuscript..

Comment 5

5. I think that the numerical results presented are not sufficient to convince the reader about the advantage of the method.

Authors reply

Agreed. We have updated the manuscript by providing the necessary elaborative information of the simulations along with their statistical assessments and complexity studies in the results section to facilitate the reader more clearly to decipher the contributions of the study. Please see the updated simulation results presented in the results section.

Comment 6 

6. The list of references is quite poor and it should be enlarged.

Authors Reply

Agreed. We have updated the manuscript by providing the exhaustive literature review in the introduction section to portray the problem statement with justification with the help of relevant, recent and reputed journal articles as suggested. Many thanks for suggesting the relevant recent articles [r1-r18] on the topic and indeed these article are very helpful to improve the technical and presentation strength of the introduction section. 

[r1] Ahmad, A., Sulaiman, M., Aljohani, A.J., Alhindi, A. and Alrabaiah, H., 2021. Design of an efficient algorithm for solution of Bratu differential equations. Ain Shams Engineering Journal, 12(2), pp.2211-2225.

[r2] Waseem, W., Sulaiman, M., Kumam, P., Shoaib, M., Raja, M.A.Z. and Islam, S., 2020. Investigation of singular ordinary differential equations by a neuroevolutionary approach. Plos one, 15(7), p.e0235829.

[r3] Zhang, Y., Lin, J., Hu, Z., Khan, N.A. and Sulaiman, M., 2021. Analysis of third-order nonlinear multi-singular Emden–Fowler equation by using the LeNN-WOA-NM algorithm. IEEE Access, 9, pp.72111-72138.

[r4] Fawad Khan, M., Bonyah, E., Alshammari, F.S., Ghufran, S.M. and Sulaiman, M., 2022. Modelling and Analysis of Virotherapy of Cancer Using an Efficient Hybrid Soft Computing Procedure. Complexity, 2022.

[r5] Rahman, I.U., Sulaiman, M., Alarfaj, F.K., Kumam, P. and Laouini, G., 2021. Investigation of Non-Linear MHD Jeffery–Hamel Blood Flow Model Using a Hybrid Metaheuristic Approach. IEEE Access, 9, pp.163214-163232.

[r6] Bukhari, A.H., Raja, M.A.Z., Sulaiman, M., Islam, S., Shoaib, M. and Kumam, P., 2020. Fractional neuro-sequential ARFIMA-LSTM for financial market forecasting. IEEE Access, 8, pp.71326-71338.

[r7] Bukhari, A.H., Sulaiman, M., Raja, M.A.Z., Islam, S., Shoaib, M. and Kumam, P., 2020. Design of a hybrid NAR-RBFs neural network for nonlinear dusty plasma system. Alexandria Engineering Journal, 59(5), pp.3325-3345.

[r8] Bukhari, A.H., Sulaiman, M., Islam, S., Shoaib, M., Kumam, P. and Raja, M.A.Z., 2020. Neuro-fuzzy modeling and prediction of summer precipitation with application to different meteorological stations. Alexandria Engineering Journal, 59(1), pp.101-116.

[r9] Guirao, J.L., Sabir, Z., Raja, M.A.Z. and Baleanu, D., 2022. Design of neuro-swarming computational solver for the fractional Bagley–Torvik mathematical model. The European Physical Journal Plus, 137(2), p.245.

[r10] Sabir, Z., Raja, M.A.Z., Baleanu, D., Cengiz, K. and Shoaib, M., 2021. Design of Gudermannian Neuroswarming to solve the singular Emden–Fowler nonlinear model numerically. Nonlinear Dynamics, 106(4), pp.3199-3214.

[r11] Iqbal, M.A., Fakhar, M.S., Kashif, S.A.R., Naeem, R. and Rasool, A., 2021. Impact of parameter control on the performance of APSO and PSO algorithms for the CSTHTS problem: An improvement in algorithmic structure and results. PloS one, 16(12), p.e0261562.

[r12] Zhang, X. and Tang, Z., 2022. Construction of computer model for enterprise green innovation by PSO-BPNN algorithm and its impact on economic performance. Plos one, 17(1), p.e0262963.

[r13] Hosseinzadeh Khonakdari, T. and Ahmadi Kamarposhti, M., 2021. Real-time detection of microgrid islanding considering sources of uncertainty using type-2 fuzzy logic and PSO algorithm. PloS one, 16(9), p.e0257830. 

[r14] Chen, F., Gao, X., Xia, X. and Xu, J., 2022. Using LSTM and PSO techniques for predicting moisture content of poplar fibers by Impulse-cyclone Drying. PloS one, 17(4), p.e0266186.

[r15] Umar, M., Sabir, Z., Raja, M.A.Z., Amin, F., Saeed, T. and Guerrero-Sanchez, Y., 2021. Integrated neuro-swarm heuristic with interior-point for nonlinear SITR model for dynamics of novel COVID-19. Alexandria Engineering Journal, 60(3), pp.2811-2824.

[r16] Sabir, Z., Raja, M.A.Z., Kamal, A., Guirao, J.L., Le, D.N., Saeed, T. and Salama, M., 2021. Neuro-Swarm heuristic using interior-point algorithm to solve a third kind of multi-singular nonlinear system. Mathematical Biosciences and Engineering, 18(5), pp.5285-5308.

[r17] AlQadi, H. and Bani-Yaghoub, M., 2022. Incorporating global dynamics to improve the accuracy of disease models: Example of a COVID-19 SIR model. PloS one, 17(4), p.e0265815.

[r18] Islam, M.Z., Othman, M.L., Abdul Wahab, N.I., Veerasamy, V., Opu, S.R., Inbamani, A. and Annamalai, V., 2021. Marine predators algorithm for solving single-objective optimal power flow. Plos one, 16(8), p.e0256050.

Authors are once again thankful to anonymous reviewer, editor in chief, associate editor and editorial staff for their time, help, efforts and support.

Best regards,

Thongchai Botmart

Corresponding author

---

## [Decision Letter · Decision Letter 1]

30 Jun 2022

PONE-D-21-38900R1A stochastic numerical approach for a class of singular singularly perturbed systemPLOS ONE

Dear Dr. Botmart,

Thank you for submitting your manuscript to PLOS ONE. After careful consideration, we feel that it has merit but does not fully meet PLOS ONE’s publication criteria as it currently stands. Therefore, we invite you to submit a revised version of the manuscript that addresses the points raised during the review process.

Two reviewers have completed the review of your manuscript. Although they suggested acceptance, during our in-house check, we noticed that your manuscript contains significant references of other works by the same authors. PLOS ONE stresses that the literature review and reference list should accurately reflect the state of the field, and opposes excessive citations. Please check and revise your manuscript again following this criteria, and submit a revised version. We cannot rule out the possibility of inviting a third reviewer.

We look forward to receiving your revised manuscript.

Kind regards,

Hanna Landenmark

Senior Editor, PLOS ONE

on behalf of

Chengming Huang

Journal Requirements:

Additional Editor Comments (if provided):

Reviewers' comments:

Reviewer's Responses to Questions

**Comments to the Author**

1. If the authors have adequately addressed your comments raised in a previous round of review and you feel that this manuscript is now acceptable for publication, you may indicate that here to bypass the “Comments to the Author” section, enter your conflict of interest statement in the “Confidential to Editor” section, and submit your "Accept" recommendation.

Reviewer #1: All comments have been addressed

Reviewer #2: All comments have been addressed

2. Is the manuscript technically sound, and do the data support the conclusions?

Reviewer #1: Yes

Reviewer #2: Yes

3. Has the statistical analysis been performed appropriately and rigorously? 

Reviewer #1: Yes

Reviewer #2: Yes

4. Have the authors made all data underlying the findings in their manuscript fully available?

Reviewer #1: Yes

Reviewer #2: No

5. Is the manuscript presented in an intelligible fashion and written in standard English?

Reviewer #1: Yes

Reviewer #2: Yes

6. Review Comments to the Author

Reviewer #1: This paper is revised very well. All comments are addressed. It may be accepted in its current form.

Reviewer #2: Now the manuscript is sutible for publication.

7. PLOS authors have the option to publish the peer review history of their article (what does this mean?). If published, this will include your full peer review and any attached files.

Reviewer #1: No

Reviewer #2: No

---

## [Author Response · Author response to Decision Letter 1]

28 Jul 2022

Author Response

Paper Title: 

A stochastic numerical approach for a class of singular singularly perturbed system 

by

Zulqurnain Sabir, Thongchai Botmart, Muhammad Asif Zahoor Raja, Wajaree weera, Fevzi Erdoğan

ID: PONE-D-21-38900R1

Journal: PLOS ONE

Reviewers' comments:

Reviewer #1: All comments have been addressed

Authors reply

Many thanks for your valuable remarks and encouraging recommendations

Reviewer #2: All comments have been addressed

Authors reply

Many thanks for your valuable remarks and encouraging recommendations

Many thanks for the 

Editorial Comments

Thank you for submitting your manuscript to PLOS ONE. After careful consideration, we feel that it has merit but does not fully meet PLOS ONE’s publication criteria as it currently stands. Therefore, we invite you to submit a revised version of the manuscript that addresses the points raised during the review process.

Two reviewers have completed the review of your manuscript. Although they suggested acceptance, during our in-house check, we noticed that your manuscript contains significant references of other works by the same authors. PLOS ONE stresses that the literature review and reference list should accurately reflect the state of the field, and opposes excessive citations. Please check and revise your manuscript again following this criteria, and submit a revised version. We cannot rule out the possibility of inviting a third reviewer.

Authors reply

First of all many thanks for both reviewer for acceptance of our manuscript in your esteemed journal after address the technical comments. 

We have updated the manuscript after addressing the editorial remarks regarding the reference cited in the literature review,; accordingly, after detailed, careful and critical review of the literature review for the justification of the problem statement, we have cited only those references accurately reflect/portray the state of the field. Additionally we have avoided the extensive citation of the references of the particular author as suggested for better understanding and readability.

Please see the introduction and reference section of the revised manuscript.

---

## [Decision Letter · Decision Letter 2]

25 Oct 2022

A stochastic numerical approach for a class of singular singularly perturbed system

PONE-D-21-38900R2

Dear Dr. Botmart,

We’re pleased to inform you that your manuscript has been judged scientifically suitable for publication and will be formally accepted for publication once it meets all outstanding technical requirements.

Kind regards,

Dragan Pamucar

Academic Editor

PLOS ONE

Additional Editor Comments (optional):

Reviewers' comments:

Reviewer's Responses to Questions

**Comments to the Author**

1. If the authors have adequately addressed your comments raised in a previous round of review and you feel that this manuscript is now acceptable for publication, you may indicate that here to bypass the “Comments to the Author” section, enter your conflict of interest statement in the “Confidential to Editor” section, and submit your "Accept" recommendation.

Reviewer #1: All comments have been addressed

2. Is the manuscript technically sound, and do the data support the conclusions?

Reviewer #1: Yes

3. Has the statistical analysis been performed appropriately and rigorously? 

Reviewer #1: Yes

4. Have the authors made all data underlying the findings in their manuscript fully available?

Reviewer #1: Yes

5. Is the manuscript presented in an intelligible fashion and written in standard English?

Reviewer #1: Yes

6. Review Comments to the Author

Reviewer #1: (No Response)

7. PLOS authors have the option to publish the peer review history of their article (what does this mean?). If published, this will include your full peer review and any attached files.

Reviewer #1: No
